# Chronic BACE-1 Inhibitor Administration in TASTPM Mice (APP KM670/671NL and PSEN1 M146V Mutation): An EEG Study

**DOI:** 10.3390/ijms21239072

**Published:** 2020-11-28

**Authors:** Susanna Lopez, Claudio Del Percio, Gianluigi Forloni, Angelisa Frasca, Wilhelmus H. Drinkenburg, Roberta Lizio, Giuseppe Noce, Raffaele Ferri, Andrea Soricelli, Fabrizio Stocchi, Laura Vacca, Règis Bordet, Jill C. Richardson, Claudio Babiloni

**Affiliations:** 1Department of Physiology and Pharmacology “V Erspamer”, Sapienza University of Rome, 00185 Rome, Italy; susanna.lopez@uniroma1.it (S.L.); claudio.delpercio@uniroma1.it (C.D.P.); 2Department of Emergency and Organ Transplantation - Nephrology, Dialysis and Transplantation Unit, Aldo Moro University of Bari, 70121 Bari, Italy; 3Department of Neuroscience, Istituto di Ricerche Farmacologiche Mario Negri IRCCS, 20156 Milan, Italy; gianluigi.forloni@marionegri.it (G.F.); angelisa.frasca@gmail.com (A.F.); 4Janssen Research and Development, Pharmaceutical Companies of J&J, B-2340 Beerse, Belgium; WDRINKEN@its.jnj.com; 5IRCCS SDN, 80143 Naples, Italy; roberta.lizio@uniroma1.it (R.L.); giuseppe.noce@uniroma1.it (G.N.); soricelli@uniparthenope.it (A.S.); 6Oasi Research Institute - IRCCS, 94018 Troina, Italy; rferri@oasi.en.it; 7Department of Motor Sciences and Healthiness, University of Naples Parthenope, 80133 Naples, Italy; 8Institute for Research and Medical Care, IRCCS San Raffaele Pisana, 00163 Rome, Italy; fabrizio.stocchi@sanraffaele.it (F.S.); laura.vacca@sanraffaele.it (L.V.); 9University of Lille, Inserm, CHU, 59000 Lille, France; regis.bordet@univ-lille2.fr; 10U1171 - Degenerative and Vascular Cognitive Disorders, 59000 Lille, France; 11GlaxoSmithKline R&D Neuroscience Therapeutic Area UK, Gunnels Wood Road, Stevenage, Hertfordshire SG1 2NY, UK; Jill.C.Richardson@gsk.com; 12San Raffaele Cassino, 03043 Cassino (FR), Italy

**Keywords:** electroencephalography (EEG), Alzheimer’s disease (AD), β-site amyloid precursor protein cleaving enzyme (BACE-1) inhibitor, TASTPM mice

## Abstract

Objective. In this exploratory study, we tested whether electroencephalographic (EEG) rhythms may reflect the effects of a chronic administration (4 weeks) of an anti-amyloid β-site amyloid precursor protein (APP) cleaving enzyme 1 inhibitor (BACE-1; ER-901356; Eisai Co., Ltd., Tokyo, Japan) in TASTPM (double mutation in APP KM670/671NL and PSEN1 M146V) producing Alzheimer’s disease (AD) amyloid neuropathology as compared to wild type (WT) mice. Methods. Ongoing EEG rhythms were recorded from a bipolar frontoparietal and two monopolar frontomedial (prelimbic) and hippocampal channels in 11 WT Vehicle, 10 WT BACE-1, 10 TASTPM Vehicle, and 11 TASTPM BACE-1 mice (males; aged 8/9 months old at the beginning of treatment). Normalized EEG power (density) was compared between the first day (Day 0) and after 4 weeks (Week 4) of the BACE-1 inhibitor (10 mg/Kg) or vehicle administration in the 4 mouse groups. Frequency and magnitude of individual EEG delta and theta frequency peaks (IDF and ITF) were considered during animal conditions of behaviorally passive and active wakefulness. Cognitive status was not tested. Results. Compared with the WT group, the TASTPM group generally showed a significantly lower reactivity in frontoparietal ITF power during the active over the passive condition (*p* < 0.05). Notably, there was no other statistically significant effect (e.g., additional electrodes, recording time, and BACE-1 inhibitor). Conclusions. The above EEG biomarkers reflected differences between the WT and TASTPM groups, but no BACE-1 inhibitor effect. The results suggest an enhanced experimental design with the use of younger mice, longer drug administrations, an effective control drug, and neuropathological amyloid markers.

## 1. Introduction

Alzheimer’s disease (AD) is the most diffuse progressive neurodegenerative disorder that affects aging [1,2,3]). This disorder causes dementia, clinically characterized by severe cognitive and psychiatric symptoms with a loss of autonomy in the activities of daily life [4,5,6]. 

Previous studies have reported differences in absolute and relative electroencephalographic (EEG) power related to a resting state eyes-closed condition in cognitively normal elderly (Nold) seniors compared with AD patients. As compared to Nold subjects, AD patients with dementia showed a higher power of widespread delta (<4 Hz) and theta (4–7 Hz) rhythms over the scalp, associated with lower power of posterior dominant alpha (8–12 Hz) rhythms and those at beta (13–20 Hz) frequencies [7,8,9,10,11,12,13,14,15]). 

An interesting issue is evaluating the extent to which ongoing EEG rhythms are abnormal in transgenic mouse models of AD, taking into account the significant differences in the structural and functional features of brain neurophysiological mechanisms underpinning the generation of ongoing EEG activity and the regulation of vigilance in the two species. 

A wealth of studies has documented abnormal EEG activities at delta and theta rhythms (i.e., hyper-synchronous activities) in APP- and PS1-mutated transgenic models showing progressive amyloidosis in the brain: (1) mice with mutations in the APP gene showed abnormal ongoing EEG rhythms [16]. (2) Mice carrying mutated human APP Swedish and PS1 genes showing fibrillogenic Aβ1-42 and amyloid plaques exhibited reduced theta (absolute power) and enhanced beta and gamma (absolute power) EEG rhythms recorded in wakefulness; however, those changes did not dependent on Aβ1-42 deposits as they did not progress over aging from 9 months of age [17]. Furthermore, APP but not PS1 single mutant mice, had similar alterations in theta, beta, and gamma EEG rhythms, while APP/PS1 (but not APP single-mutant) mice had high insoluble Aβ1-42/40 levels and core brain amyloid plaques at 13 months of age [17]. (3) APP Swedish and PS1 mutant mice showed increased EEG (absolute power) rhythms at a large frequency band beyond delta rhythms [18,19]. (4) The second generation of AD mouse models as triple transgenic mice (i.e., triple transgenic mice express low levels of mutant human APP, tau, and presenilin-1) presented abnormal EEG rhythms [20,21,22]. (5) The same was true in humanized mice containing functional human cells or tissues [23,24]. (6) Triple transgenic mice (i.e., 3×Tg and PLB1 triple) over controls were characterized by a decrease of theta EEG rhythms during wakefulness or cognitive tasks [21,22] and an increase of delta EEG rhythms during wakefulness and REM sleep [20]. (7) Triple-transgenic AD mice challenged with the potassium channel blocker 4-aminopyridine showed reduced theta EEG rhythms compared with 4-aminopyridine-treated control mice [22]. (8) In rodents, abnormalities in delta EEG rhythms were related to impairments in long-term and short-term hippocampal plasticity and cognitive deficits in recognition memory and spatial learning [20]. (9) As compared to mice humanized to apolipoprotein APOE3, mice humanized to APOE4 showed abnormal beta EEG rhythms during olfactory tasks [25]. (10) In transgenic mice, the activation of basal forebrain cholinergic neurons, traced by humanized Renilla green fluorescent protein (hrGFP), significantly and lastingly decreased the delta EEG rhythms, produced low-delta non-rapid eye movement sleep, and slightly increased wakefulness, whereas the inhibition of basal forebrain cholinergic neurons significantly increased delta EEG rhythms and slightly decreased wakefulness [24].

In the European Innovative Medicine Initiative (IMI) project entitled “PharmaCog” (2010–2015), ongoing EEG rhythms in wakefulness were recorded in C57BL/6 mice with and without genetic mutations of APP or APP and PS1 genes (for the sake of simplicity, we termed C57BL/6 mice without those genetic mutations as “wild type”, WT). The data analysis design was focused on the comparison of ongoing EEG activity in two behavioral conditions in the mice, namely a condition of immobility or minimal movements in wakefulness (passive condition) and a condition of exploratory movements related to a greater level of brain arousal associated with attention, sensory, and cognitive-motor processes (active condition). In the previous three PharmaCog studies, the active over the passive condition was characterized by significant changes in ongoing EEG rhythms in WT and transgenic mice [26,27,28]. 

Keeping in mind the above results, are ongoing EEG rhythms recorded in wakefulness sensitive to the effects of drugs reducing the cerebral accumulation of Aβ peptide in transgenic mouse models of AD? To our knowledge, this is a relevant but still open issue for an application in preclinical pathways of AD-modifying drug discovery. In the present study, we report the original results of a fourth PharmaCog EEG study aimed at testing the hypothesis that an anti-amyloid inhibitor of the β-site APP-cleaving enzyme 1 (BACE-1; ER-901356; Eisai Co, Ltd., Tokyo, Japan), given for 4 weeks over the vehicle may affect ongoing EEG rhythms in 8–9 month old TASTPM mice, when compared to WT littermates.

## 2. Results

### 2.1. Normalized EEG Power (Density) during Active and Passive Conditions in WT and TASTPM Mice

Figure 1 shows the grand average of the normalized EEG power (density) in the WT Vehicle (*n* = 11), WT BACE-1 (*n* = 10), TASTPM Vehicle (*n* = 8), and TASTPM BACE-1 (*n* = 7) at the bipolar frontoparietal channel at Day 0 and Week 4.

As expected, all four groups of mice were characterized by differences in the normalized EEG power (density) during the passive and active conditions at both EEG recording time points. During the passive condition, the normalized EEG power peak was observed at 1–6 Hz (i.e., delta range; IDF peak), with EEG power greater than that observed at the theta range (i.e., 6.5–10 Hz). Conversely, the normalized EEG power peak was observed at 6.5–10 Hz during the active condition, with EEG power greater than that observed at the delta range.

The same results were obtained when frontomedial prelimbic (Figure 2) and hippocampal (Figure 3) electrodes were considered.

### 2.2. Control Analysis of High Frequency Monopolar Parietal EEG Signals

To cross-validate the behavioral classification in the active and passive conditions, we analyzed the parietal electrophysiological signal at very high frequencies mainly reflecting electromyographic (EMG) activity generated from neck and back muscles relatively close to the parietal electrode. The hypothesis was that such EMG-like signal was greater in power (density) in all groups of mice during the active than the passive condition (*p* < 0.05). To obtain that EMG-like activity, we applied the same normalization procedure as described above from 1 to 240 Hz and calculated the average value of the normalized EEG power density at the parietal electrode between 160 and 240 Hz (excluding frequency bins between 190 and 210 Hz to avoid harmonics of the net current). 

Figure 4 illustrates the individual values of that EMG-like power (density) for the active and the passive condition in the 4 groups of mice (WT Vehicle, WT BACE-1, TASTPM Vehicle, TASTPM BACE-1) at Day 0 and Week 4. As expected, the normalized EMG-like power was higher during the active as compared to the passive condition. This effect was statistically confirmed by a 2-way ANOVA having the normalized EMG-like power between 160 and 240 Hz at the parietal electrode as a dependent variable (*p* <0.05). The ANOVA factors were Group (WT, TASTPM) and Condition (active, passive). The only statistically significant effect was a main Condition effect (F (1, 60) = 15.1, *p* < 0.0005) with higher values of the dependent variable during the active than the passive condition.

### 2.3. Individual Values of the Normalized EEG Power at the Frontoparietal, Frontomedial Prelimbic, and Hippocampal Electrodes 

Figure 5 (left column) illustrates the individual values of the frontoparietal normalized EEG power (density) for the WT Vehicle, WT BACE-1, TASTPM Vehicle, and TASTPM BACE-1 mouse groups at the frequencies of interest (IDF, ITF, beta, and ITF/IDF), the two behavioral conditions (active and passive), and the two recording times (day 0 and week 4). Two WT Vehicle, two WT BACE-1, and one TASTPM BACE-1 mice were excluded from the following analyses as they showed outlier values according to the Grubbs test (arbitrary threshold of *p* < 0.01). 

The same figure illustrates the individual values of the frontomedial prelimbic and hippocampal (Figure 5, middle and right column, respectively) normalized EEG power (density) for the four mouse groups. No outlier values were detected according to the Grubbs test (arbitrary threshold of *p* < 0.01). 

### 2.4. Results of the Main Statistical Analysis

No statistically significant effect (*p* > 0.05) was observed in the results of the ANOVA testing EEG power (density) at the frontoparietal, frontomedial prelimbic, and hippocampal channels as dependent variables, in relation to the group (WT and TASTPM; independent variable), condition (active and passive), band (IDF, ITF, and beta), treatment (Vehicle and BACE-1; independent variable), and time (day 0 and week 4) factors.

In contrast, a statistically significant effect was observed in the results of the ANOVAs testing the power ratio between ITF and IDF (ITF/IDF) at the frontoparietal channel as a dependent variable in relation to the group (WT and TASTPM; independent variable), condition (active and passive), treatment (Vehicle and BACE-1; independent variable), and time (day 0 and week 4) factors (*p* < 0.05). Specifically, there was a significant main effect of the condition factor (F (1, 52) = 84.1, *p* < 0.001) and a significant interaction of the condition X group factors (F (1, 52) = 5.0, *p* < 0.05). Duncan post-hoc test indicated a statistically significant difference for the ITF/IDF normalized frontoparietal EEG power between the WT and TASTPM mice during the active condition (*p* < 0.05). No effect of the treatment or time factor was observed (*p* < 0.05). Figure 6 illustrates the above-mentioned effects.

No statistically significant effect was observed in the results of the ANOVA testing the power ratio between ITF and IDF (ITF/IDF) at the frontomedial prelimbic and hippocampal channels as dependent variables in relation to the group (WT and TASTPM; independent variable), condition (active and passive), treatment (Vehicle and BACE-1; independent variable), and time (Day 0 and Week 4) factors (*p* > 0.05).

### 2.5. Control Analysis

The main results of the present study showed that the BACE-1 inhibitor ER-901356 treatment lasting four weeks did not affect EEG rhythms in WT and TASTPM mice (PS1 and APP mutations). 

In this second revision of the manuscript, we reported the results of a control experiment aimed at testing the sensitivity of the present EEG markers to Donepezil, namely a reversible Acetylcholinesterase inhibitor with low toxicity, licensed for the symptomatic treatment of cognitive deficits in AD patients [29]. In previous studies, its chronic administration produced some beneficial effects on cognitive functions in AD patients with mild, moderate, or severe dementia [30]. It also slowed down the progression of the hippocampal atrophy in AD patients with dementia as compared to untreated ones [31]. Furthermore, a single dose of Donepezil induced activating effects on EEG data recorded in both WT mice [32] and rats [33].

In the present control experiment, we hypothesized that, in the on-going EEG activity recorded in WT mice, the ITF/IDF normalized frontoparietal EEG power may show (1) a statistically significant difference between the passive and active conditions after the administration of both vehicle and Donepezil and (2) greater values of that difference in the active condition after the Donepezil than the vehicle administration (*p* < 0.05). 

The data of this control experiment derived from the Work Package 2 activities of the “PharmaCog” project, which included preclinical studies performed in rodents to test some biomarkers of symptomatic treatments for AD. From that Work Package, we had access to the on-going EEG data recorded in the following rodents: (1) 5 WT male mice 18 months old receiving a single dose of vehicle by i.p. injection and (2) 5 WT male mice 18 months old receiving a single dose of Donepezil by i.p. injection (1 mg/kg; 5 mice, aged 18 months). In addition, we had also access to the on-going EEG data recorded in 9 WT male mice 12 months old receiving no pharmacological intervention. 

All experimental procedures (i.e., animal management, data recording and analysis, and the computation of the power ratio between ITF and IDF—ITF/IDF—at the frontoparietal channel as a dependent variable) of the control experiment were generally those of the main experiment performed in WT and TASTPM mice receiving vehicle or BACE-1 inhibitor ER-901356 treatment (see Methods section). As a difference, the on-going EEG data in the mice of the control experiment was performed only from the frontoparietal electrodes. 

In the WT mice receiving the vehicle or Donepezil, the EEG period of interest ranged between 85 and 95 min following i.p. injection according to the pharmacodynamics properties of that drug [34]. The normalized ITF/IDF EEG spectral power density at the frontoparietal derivation was analyzed for the artifact-free EEG activity recorded during the passive and active conditions, following the procedures used for the main experiment. The statistical analysis of the normalized ITF/IDF EEG spectral power density was performed by a nonparametric test due to the relatively small number of mice (Wilcoxon test; *p*< 0.05). In the WT mice receiving the vehicle or Donepezil, the contrasts of interest were (1) between the passive and active condition within each intervention and (2) between the vehicle and Donepezil interventions for the active condition. 

The main results of the control experiment are represented in Figure 7. In the WT mice receiving the vehicle or Donepezil (18 months old), the normalized ITF/IDF EEG spectral power density at the frontoparietal derivation was significantly greater in the active than the passive condition after the administration of both the vehicle and Donepezil (*p* < 0.05). Furthermore, in the active condition, such power density marker was significantly greater after the administration of the Donepezil than the vehicle (*p* < 0.05). In the mice receiving no intervention with vehicle or Donepezil (12 months old), the normalized ITF/IDF EEG spectral power density at the frontoparietal derivation was also significantly greater in the active than the passive condition (*p* < 0.05). In conclusion, the results of the control experiment suggest that in the WT mice, the normalized ITF/IDF EEG spectral power density at the frontoparietal derivation used in the main experiment (BACE-1 inhibitor vs. vehicle) was sensitive to a single dose of Donepezil over the vehicle. 

## 3. Discussion

Here, we tested whether ongoing EEG rhythms at (individual) delta and theta frequencies may be able to reflect neurophysiological effects of a chronic treatment affecting insoluble amyloid β in the brain in an AD mouse model, i.e., the TASTPM mice (age of 8–9 months). Specifically, we administered a well-known inhibitor of the β-site APP-cleaving enzyme 1 (BACE-1; ER-901356; Eisai Co, Ltd., Tokyo, Japan) over a vehicle for 4 weeks in both TASTPM and WT mice (cross-sectional design) and used the normalized EEG power (density) as a measure of brain neural synchronization in behavioral passive and active conditions classified in wakefulness. To our knowledge, this is the first study evaluating spectral EEG markers in TASTPM mice in relation to a chronic administration of a BACE-1 inhibitor.

### 3.1. Poor “Reactivity” of Ongoing EEG Rhythms in TASTPM Mice during Exploratory Movements

Results showed that both WT and TASTPM mice were generally characterized by specific frequency features of on-going EEG rhythms during the active and passive behavioral conditions. Specifically, they showed a dominant individual EEG power peak at 1-6 Hz (i.e., delta range; IDF peak) during the passive condition. During the active condition, such dominant IDF power decreased together with an increase in the EEG power peaking at 6.5-10 Hz (i.e., theta range; ITF peak). This “reactivity” of the IDF and ITF power during the active condition was observed at all cortical and subcortical electrodes of the montage, located at frontoparietal, frontomedial prelimbic, and hippocampal CA3 regions. It was observed in both Day 0 and Week 4 recordings and in all four groups of mice. 

These above results motivated the use of an index reflecting this “reactivity” of ongoing EEG rhythms < 10 Hz as the ratio of the EEG power between ITF and IDF (i.e., ITF/IDF). The statistical analysis with this new index showed that the frontoparietal ITF/IDF power was affected by the genotype (WT and TASTPM) and behavioral condition (active and passive) as well. As compared to the WT mice, the TASTPM mice were characterized by lower ITF/IDF power during the active state, thus confirming the poor “reactivity” of those EEG rhythms during exploratory movements in mouse mutants expressing an accumulation of amyloid β in the brain (i.e., mutations of PS1 and APP). 

The mentioned results confirmed previous EEG evidence showing increased brain theta power and/or reduced delta power in mice during overt movements as compared to quiet wakefulness [16,18,19,27,35,36,37,38]. They are also in agreement with previous evidence of the PharmaCog project carried out not only in TASTPM mice [28] but also in mice with the only mutation of APP, namely the PDAPP mice [27]. Therefore, it can be speculated that the mutation of APP may be sufficient to produce the documented poor “reactivity” of those EEG rhythms when mice are performing exploratory movements. 

The poor condition-dependent EEG “reactivity” in the present 8–9-month old TASTPM mice may depend on several factors. Previous studies have shown that at the age of 6–24 months, TASTPM mice are characterized by relevant AD manifestations, such as amyloid β deposition with dystrophic neurites in the cerebral cortex, hippocampus, and thalamus, in association with cognitive deficits [39,40]. In those previous experiments performed in TASTPM mice, brain amyloidosis was also related to hypometabolism, neuroinflammation, and neurodegeneration in the cerebral cortex, hippocampus, basal ganglia, and thalamus [41]. Of note, those cerebral regions are important nodes of neural networks regulating ongoing EEG rhythms and brain arousal in wakefulness [42,43,44,45,46]. 

Taking together the previous and the present data, we speculate that even in the relatively young TASTPM mice of the present study (8–9 months of age), the poor increase in frontoparietal theta power during the cage exploration may reflect the impact of AD neuropathology on thalamus-cortical neurophysiological mechanisms underpinning cortical arousal, vigilance, and maybe cognitive-motor functions.

### 3.2. No effect of the BACE-1 Inhibitor on Ongoing EEG Rhythms in TASTPM Mice

The novel and most interesting findings of the present study showed that the IDF, ITF, and ITF/IDF power showed no significant effect of the BACE-1 inhibitor over the vehicle treatment at any mouse group (i.e., WT and TASTPM at the age of 8–9 months) or cortical and hippocampal electrodes. In other words, the administration of the BACE-1 inhibitor for 4 weeks in the TASTPM mice did not recover the poor “reactivity” of frontoparietal, frontomedial, and hippocampal ITF and IDF power during the active condition. 

These “negative” findings complement those obtained in parallel experiments carried out in the PharmaCog project, which were reported in the PharmaCog Final Report Deliverable 6.3-6.4 released to European Commission in December 2015 (“Final Report”. Available online: https://www.imi.europa.eu/projects-results/project-factsheets/pharma-cog (accessed on 25 November 2020)). In those PharmaCog experiments, other groups of 8–9-month old WT and TASTPM mice received the same dose of BACE-1 inhibitor (10 mg/kg) against vehicle for 12 weeks with the hypothesis that the drug may induce beneficial effects on the brain at structural and molecular levels, as revealed by magnetic resonance imaging (MRI). Unfortunately, the results did not confirm that hypothesis. When compared to the WT mice, the TASTPM mice generally showed lower volume in hippocampus, striatum, and a thinning of the entorhinal cortex. However, no effect of the treatment using the BACE-1 inhibitor was observed over time (PharmaCog Final Report Deliverable D 6.3–6.4, December 2015). In the same line, the diffusion tensor imaging (DTI) analysis of the MRIs revealed general lower values in the fractional anisotropy of the external capsule and hippocampus in the TASTPM in relation to the WT mice, but again no effect of the BACE-1 inhibitor over time was observed (PharmaCog Final Report deliverable D 6.3–6.4, December 2015). Furthermore, the magnetic resonance spectroscopy (MRS) analysis revealed general lower values in GABA and glutamate and a trend towards lower values in myo-inositol in the hippocampus of the TASTPM over the WT mice, but again no effect of the BACE-1 inhibitor over time was observed (PharmaCog Final Report deliverable D 6.3–6.4, December 2015). 

Unfortunately, we do not have a conclusive explanation about the negligible effects of the chronic administration of the present BACE-1 inhibitor (ER-901356; Eisai Co., Ltd.) on brain structural integrity and functions including neurophysiological oscillatory mechanisms underlying ongoing EEG rhythms measured in the present TASTPM mice during passive and active behavioral conditions. We can just speculate about that explanation in the following paragraphs. 

As a first line of speculation, the present BACE-1 inhibitor may not recover ongoing EEG rhythms in TASTPM mice be due to late age of the rodents and short administration period. It should be considered that BACE-1 inhibitors mainly act on the formation of neurotoxic Aβ1-42 oligomers, which occurs much before the deposition of amyloid β plaques occurring at 6–9 months of age. Therefore, the administration of BACE-1 inhibitors at the age of 8–9 months in the current experiments could be late to mitigate (1) the initial effects of insoluble Aβ1-42 on brain parenchyma and (2) the formation of amyloid hard plaques occurring from 6 months of age in TASTPM mice (Howlett et al., 2004; Howlett et al., 2008). In line with this speculation, previous evidence showing beneficial effects of BACE-1 inhibitors on neuronal functions and cognitive performance was obtained in young mouse mutants giving the drug before the formation of the amyloid β plaques. Specifically, Chang et al. (2010) showed that 5.5-month old Tg2576 mice (overexpressing a mutant form of APP, isoform 695, with the Swedish mutation, KM670/671NL) continuously infused with the BACE-1 inhibitor for 6 months were characterized by a reduced accumulation of the brain and plasma amyloid β by about 60–80% with reference to the control mice. In three separate experiments, AD mouse models ranging from 5.5 to 9 months of age showed beneficial effects in cognitive performance in the Morris’ Water Maze when treated with a BACE-1 inhibitor from about 4 to 7 months. Noteworthy, shorter treatment periods or starting that treatment at an older age (16 months) failed to show those beneficial effects on cognitive performances, despite a significant reduction of brain Aβ accumulation [47]. 

Concerning the neurophysiological mechanisms underlying the beneficial effects reported in the studies mentioned above, the chronic administration with the BACE-1 inhibitor restored the impaired synchronization of single neurons around amyloid plaques and in neuronal populations distributed in different brain regions [48,49]. It was speculated that the effects were due to an improvement in the neuronal crosstalk in several areas of the brain (occipital, somatosensory, motor, and frontal regions; [50]), reminiscent of the default network activity observed in humans [49,51]. In those encouraging previous experiments, the improvement in the neuronal function due to the BACE-1 inhibitor treatment could occur despite high amyloid plaque load, probably in relation to the lack of a significant brain neurodegeneration.

In the PharmaCog scientific program, the BACE-1 inhibitor was not administered in mice younger than 8–9 month old to avoid the possible confounding adverse effects of an early administration of the drug on brain development [52,53,54]. Indeed, previous studies showed that in BACE-1 knock-out mice, some development deficits were found, such as errors in axon targeting [55,56,57], reduced axon myelination [58,59,60], and deficits in synaptic transmission and plasticity at the hippocampal Schaffer collateral to CA1 synapses [61]. These previous findings are not surprising as BACE-1 inhibitor may serve several physiological functions supporting synaptic transmission [62,63] and accurate axon guidance [53]. Furthermore, it contributes to the cleaving and release of the neurotrophic factor neuregulin-1 (NRG-1; [15,64]). 

As a second line of speculation, the present BACE-1 inhibitor may not recover ongoing EEG rhythms in TASTPM mice due to the complex relationship between their beneficial effects on amyloid β accumulation in the brain and general brain functions [65,66]. The mere reduction of circulating Aβ1-42 oligomers may not affect the complex physiological dysregulations at the basis of cognitive deficits in TASTPM mice. Future studies may investigate the relationships among the administration of the BACE-1 inhibitor and in-vivo measures of AD-like neuropathology (e.g., amyloid β, tau, neuroinflammation), the present EEG markers, and cognitive functions. 

Keeping in mind the above data and considerations, the present EEG markers might be useful in the evaluation of the effects of BACE-1 inhibitor treatment on neurophysiological mechanisms regulating the vigilance state in a prevention framework. Indeed, the alteration of EEG delta and theta rhythms in the TASTPM mice of this study did not merely reflect the changes in circulating insoluble amyloid β. It can be speculated that a drug-related improvement in those EEG markers may predict significant beneficial effects of the treatment on brain structure and functions including cognitive status. 

### 3.3. Methodological Limitations of the Study

The results of this exploratory study should be considered as preliminary due to the following methodological limitations. 

Firstly, a limited number of mice was available for the present experiments. As mentioned above, the PharmaCog project planned and developed several parallel exploratory experiments on the effects of the BACE-1 inhibitor ER-901356 on MRI and EEG biomarkers in TASTPM mice. Given the exploratory nature of the experiments and obvious ethical reasons about the number of animals sacrificed, those parallel studies were designed with 12 mice for any TASTPM and WT group. This number was based on preliminary data of the big Pharma companies participating in the PharmaCog project (e.g., GlaxoSmithKline, Janssen, Lundbeck, Eisai, etc., see “Who are the PharmaCog partners?”. Available online: https://www.alzheimer-europe.org/Research/PharmaCog/Who-are-the-PharmaCog-partners (accessed on 25 November 2020)). For the same reasons, only two levels of treatment were designed (i.e., vehicle and BACE-1 inhibitor ER-901356), and we decided to focus on male mice to avoid the confounding effects of the estrous ovarian cycle on EEG recordings [67] at an early stage of the research (this limitation is significant as the amyloid pathology may develop more rapidly and severely in female than male mice; [68]). 

Notably, some dropouts for mortality or EEG technical failures or artifactual dataset reduced the number of mice in some statistical contrasts. As mentioned above, during the 4 weeks of the experiment, two mice of the TASTPM Vehicle group and four mice of the TASTPM BACE-1 inhibitor group died, whereas no WT mouse died during the experiments. These results suggest that the dosage of the BACE-1 inhibitor ER-901356 and the chronic implantation of the intracranial electrodes for 4 weeks were well tolerated by the WT but not the TASTPM mice under the present experimental conditions. Those detrimental effects in the TASTPM may be due to the interaction among the TASTPM genotyping, the BACE-1 inhibitor ER-901356, and the chronic implantation of the intracranial electrodes for 4 weeks. Based on the present experimental design, it is not possible to disentangle the specific detrimental effects of the BACE-1 inhibitor ER-901356 (18% of mortality in a linear system) and the chronic electrode implantation (18% of mortality in a linear system) in the TASTPM mice, as they may be nonlinear. The tolerability of the BACE-1 inhibitor ER-901356 is clearly a methodological aspect very important for the translational perspective and should be further investigated in future preclinical EEG studies.

From the statistical point of view, the impact of the present dropouts due to the mortality and EEG failures may be moderate. The mouse groups had ≥ 7 animals and the lack of the effects of the BACE-1 inhibitor ER-901356 on EEG variables may not be due to an insufficient power as the mean values of EEG variables were very similar between the TASTPM Vehicle and BACE-1 inhibitor groups and between the WT Vehicle and BACE-1 inhibitor groups (see Figure 5). Considering the present mortality rate in the TASTPM mice and the EEG recordings rejected due to recording failure or artifacts, a statistical analysis with *p* = 0.05 and 0.8 of desired power would require a minimum number of 15 mice per group. As the present experiments were performed during the PharmaCog project in the years 2012–2015, we cannot extend them with new mice, drugs, and EEG recordings to date.

Secondly, mouse behavior was qualitatively rated as active vs. passive based on a harmonized visual rating protocol. In that protocol, movement velocity and extension were not considered. Therefore, the reported EEG differences in the two mouse groups (i.e., WT and TASTPM) might partially depend on different quantitative motor features in the active condition. This limitation is relevant as hippocampal theta rhythms might reflect some features of movements [41,69,70]. To reduce this variability, we performed a control analysis estimating the muscular activity from the electrophysiological signal at high frequencies (> 150 Hz) recorded from the monopolar parietal electrode, which may be considered to reflect the EMG activity of neck and should mouse muscles at those high frequencies.

Thirdly, TASTPM mice typically show AD-related amyloid pathology and memory deficits but not the intracellular tangle pathology and the important neural loss in the brain-characterizing patients with dementia due to AD [39,40]. Therefore, the interpretation of the current findings should consider that some potential effects of the BACE-1 inhibitor ER-901356 on the AD-related amyloid-tauopathy-neurodegenerative cascade may be missing in the present experiments using TASTPM mice. The TASTPM model may mainly enlighten about the earlier (characterized by the brain amyloidosis) rather than the late stages of the AD disease progression (characterized by the neurodegeneration). In this line, as cortical EEG activities strictly depend on the structural integrity of thalamocortical and corticothalamic systems, EEG techniques may be especially sensitive to the effects of pharmacological treatments given to mouse mutants showing intracellular tangle pathology and cortical neurodegeneration [71]. This aspect should be taken into account in the design of the drug discovery pathways and the interpretation of EEG results in future EEG investigations in mouse models of AD.

Fourthly, as mentioned above, the dosage of 10 mg/kg for the BACE-1 inhibitor ER-901356 administered to the present mice was defined by two preliminary PD/PK experiments carried out in rats by the Eisai Unit. Results of those preliminary experiments [72] consistently suggested that such a dosage could reduce Aβ levels in the rodent brain. However, we do not know if this effect may reduce the brain amyloid neuritic plaques after 4 weeks of administration and may explain the present EEG abnormalities in the TASTPM mice. This is a hypothesis for future studies.

Fifthly, the current methodology cannot substitute other classical neurophysiological methodologies applied in mice, namely long EEG recordings investigating the sleep–wake cycle, the experimental inoculation of stress or anxiety, and new technologies of virtual reality to simulate spatial navigation in animals during EEG recordings.

## 4. Methods

### 4.1. Animals

In the present study, 11 C57BL/6 (for the sake of simplicity, wild type, WT) vehicle, 10 WT BACE-1, 10 TASTPM Vehicle, and 11 TASTPM BACE-1 mice (males; aged 8/9 months old at the beginning of treatment) were used for the final EEG data analysis (see Table 1). The EEG data were collected by researchers of Mario Negri Institute for Pharmacological Research of Milan (Italy). Of note, the original experimental design planned 12 mice for each group. However, after 4 weeks of chronic administration with the vehicle/BACE-1 inhibitor, two mice of the TASTPM Vehicle group and four mice of the TASTPM BACE-1 group died, whereas no WT mice died during the experiments.

In the enrollment of the mice, the inclusion criteria were a general health status at the visual inspection, the manifestation of physiological and instinctual behaviors, the usage of nesting material, a normal body weight, and a regular coat. The exclusion criteria were the manifestation of hunched or other abnormal postures, the loss of the body weight during the study, and the presence of body wounds.

All experimental procedures involving mice and their care were conducted in line with the institutional guidelines, in strict conformity with national and international laws and policies (European Economic Community, EEC, Council Directive 86/609, OJ L 358, 1, 12 December, 1987; U.S. National Research Council, 1996, Guide for the Care and Use of Laboratory Animals). The respect of these guidelines was carefully controlled by the members of the Work Package 8 (WP8) of European IMI PharmaCog project (https://www.imi.europa.eu/projects-results/project-factsheets/pharma-cog), devoted to the ethics of research in animals and human beings.

### 4.2. Pre-Surgery (3 Weeks)

For at least 3 weeks before surgery, the mice were acclimatized. They were housed at a constant temperature (18 °C–22 °C) and relative humidity (55–65%) under a standard 12-h light/dark cycle (light-on hemicycle typically spanning from 6:00 a.m. to 6:00 p.m.) with free access to food and water. After surgery, the animals were housed in individual cages at the same conditions (typical cage size was 45 cm (length) × 24 cm (width) × 20 cm (height)). Light intensity was 90–110 lx in the room, 60 lx in the cage during the light period, and less than 1 lx during the dark period. Gentle handling for about 5–10 min was applied daily to reduce the potential stress due to housing and experimenters. Such stress was evaluated continuously along with all the duration of the experiments by veterinary experts of each center. These experts tested animal muscle relaxation and standard behavioral indices of stress in freely behaving mice (i.e., preservation of exploratory movements in the cage, preservation of instinctual activities such as drinking and eating, and body weight across pre- and post-surgical days).

### 4.3. Surgery

EEG electrodes were implanted after anesthesia performed by the inhalation of isoflurane (5%) or equithesin, pentobarbital (1%), and chloral hydrate (+4%) 3.5 mL/kg). The mice were also treated with systemic analgesics and antibiotics in line with local guidelines on surgical care. Of note, analgesics were always associated with anesthetics to avoid pain during the surgical procedure. Chloral hydrate is no longer suggested in current preclinical research.

EEG recordings were performed by a tethered GRASS system. Stainless steel insulated surface epidural electrodes were used as exploring contacts at the frontal and parietal sites (model E363/20 with a diameter of 0.56 mm, 0.022”; Plastics One, VA, USA). Two intracerebral electrodes were placed in the frontomedial prelimbic cortex and in the hippocampus (CA3 region). Another intracerebral electrode, as a reference contact, was implanted in the cerebellum. Finally, the ground electrode was implanted in the temporal bone without the removal of the muscles (model E363/1 with a diameter of 0.280 mm, 0.011”, Plastics One, VA, USA). 

The above electrodes were fixed to the skull with dental cement. EEG signals were transmitted through a plastic electrode pedestal and a connector cable to the GRASS amplifier with a maximal cable length of 50 cm.

### 4.4. Quiet Post-Surgery Period (1 Week)

The mice were treated with systemic analgesics for pain relief and antibiotics to avoid inflammatory processes during a standard post-surgical period of one week. In that period, the animals neither received handling treatment nor EEG recordings.

### 4.5. Handling Post-Surgery Period (1 Week)

In the week after the above quiet post-surgery period, the EEG activity was not recorded, but gentle handling was applied for about 2–5 min daily, and the animals were gently plugged and unplugged several times (for wired systems only) across that week to familiarize them with the procedure of EEG recording. 

### 4.6. Treatment

All mice were administered daily for 4 weeks with a dose of 10 mg/kg (milligram per kilogram) of the BACE-1 inhibitor (ER-901356; Eisai Co., Ltd., Tokyo, Japan) or with a dose of vehicle through tube feeding. Such a dose was determined based on two preliminary pharmacokinetic/dynamic (PD/PK) experiments carried out in rats by the Eisai researchers. The results were reported in a publication by the Eisai Unit [72] and in the official deliverable 6.5 of the PharmaCog project, released to (and accepted by) the European Commission’s Seventh Framework Program (FP7)-IMI Board in 2015. Here, we summarized the results of those experiments for readers’ convenience. 

In the two preliminary experiments, the BACE-1 inhibitor ER-901356 was dosed at 3 and 30 mg/kg. In the first experiment, the rats were sacrificed at 2, 4, 6, or 8 h post dose (*n* = 2/3 rats in each group at each time point). In the second experiment, the rats were sacrificed at 0.25, 0.6, 1, 2, 4, 6, or 24 h post dose (*n* = 2–3 rats in each group and each time point). Brain, cerebrospinal (CSF), and plasma samples were collected under anesthesia (sodium pentobarbital 100 mg intra-peritoneal, i.p.). For both experiments, the Aβ1-40 levels in the brain, CSF, and plasma were measured using enzyme-linked immunosorbent assay (ELISA) kits. Specifically, the circulating Aβ1-40 levels in the brain were extracted using 0.2% diethylamine (DEA)/50 mM NaCl. The brain extract was diluted with the Tris base buffer, while the CSF and plasma were diluted with the ELISA dilution buffer. Drug concentrations were measured by liquid chromatography-mass spectrometry (LC/MS/MS).

In the first experiments, the results showed that the BACE-1 inhibitor ER-901356 determined a dose-dependent reduction in the brain and CSF Aβ1-40. The maximal reduction was found between 6 and 8 h with Aβ1-40 levels appearing to start returning to pre-dose levels after that time point (mean values, *n* = 2–3 at each time point for each group of rats).

In the second experiments, the results also showed that the BACE-1 inhibitor ER-901356 induced a dose-dependent reduction in the brain and CSF Aβ1-40. The maximal reduction occurred at 4 h with Aβ1-40 levels returning to baseline by 24 h. 

### 4.7. EEG Recordings

The planned EEG experiments were performed during both the dark and light phases of the experimental days. During the EEG recording period, the mice received no handling treatment. The EEG recordings started after the second hour of the beginning of light or darkness. The sampling frequency of the EEG recordings was 1600 Hz with anti-aliasing band-pass analog filters. No notch filter was used during the EEG recordings. 

Table 2 summarizes the time flow of the treatments and procedures adopted in the experimental sessions (days are referred to the surgical event for the implantation of the electrodes).

### 4.8. Determination of the Behavioral Mode

An essential step of the data analysis was the classification of animal behavior during the EEG recordings. An expert (A.F.) blinded to the genotype of the mice and not involved in the EEG data analysis performed the behavioral classification. The behavioral classification was performed by S.L. and A.F. They used the visual inspection (i.e., video of the mice) to classify EEG recording epochs lasting 10 s into some behavioral classes. The video was available for all mice. 

The animal behavior was classified into “active” and “passive” conditions in wakefulness based on the following definitions: (1)Active behavior (condition). Animals performed overt exploratory movements in the cage for most of the given epoch. The exploratory movements were characterized by ample displacements of body parts such as trunk, head, and forelimbs. They had not to be confounded with instinctive activities (vide infra).(2)Passive behavior (condition). Animals performed no (i.e., substantial immobility) or small movements of the trunk, head, and forelimbs. The maximal duration of the immobility lasted 10 s. The maximal duration of small movements of the trunk, head, and forelimbs lasted 10 s. These criteria were expected to minimize the risk that “passive condition” be misclassified as sleep and vice-versa.

Particular attention was devoted to distinguishing the “active” and “passive” conditions as compared to other behavioral states based on the following definitions: (1)Behavioral sleep state. Each epoch of the sleep state, as behaviorally defined, corresponded to the immobility of the animals for the entire period of observation (inferred by visual analysis) and during longer periods of immobility lasting several minutes with signs of muscle relaxation. As mentioned above, attention was devoted to avoiding misinterpretation of the passive condition and sleep state.(2)Instinctive behavior. This behavioral class was detected when the animal showed movements such as cleaning, drinking, eating, etc., for most of the period (inferred by visual analysis). As mentioned above, special attention was paid not to include these epochs in the active behavior epochs.(3)Undefined. Each epoch classified as undefined showed a mix of the other behavioral classes or lack of clarity about the behavioral situation of the animal. Such epochs were excluded from the analysis.

According to the PharmaCog guidelines, the experiment should not use EEG data to classify the epochs to avoid circular logic. Based on the above analysis of the behavioral states, the first 5 min of artifact-free EEG epochs classified as the active condition were selected for the EEG data analysis. The same procedure of selection was followed in the selection of the artifact-free EEG epochs in the passive condition. 

### 4.9. Preliminary EEG Data Analysis

The behavioral epochs of the active and passive conditions were segmented off-line in consecutive epochs lasting 2 s each. The 2 s EEG epochs with muscle, EEG, electrocardiographic (EKG), instrumental or other artifacts (no epileptic-like EEG activity) were excluded from the EEG data analysis. For each mouse, the EEG data analysis was performed by experimenters blinded to its genotype, recording time, and the treatment administered.

As mentioned above, ongoing EEG data were recorded with electrodes placed in frontal, parietal, frontomedial prelimbic, and hippocampal (CA3) regions. To align with the previous preclinical EEG studies of the PharmaCog project [26,27,28], the recorded EEG data were re-referenced to obtain bipolar frontoparietal EEG signals by a standard procedure consisting of the subtraction of the EEG signal recorded by the monopolar parietal channel from the EEG signal recorded by the frontal monopolar channel. The bipolar frontoparietal EEG signals were then inspected together with the monopolar prelimbic and hippocampal EEG signals for the artefact rejection and used as an input for the subsequent statistical analysis of EEG rhythms. At the end of the preliminary analysis, we obtained the needed sets of artifact-free EEG epochs at all electrodes, namely frontoparietal, frontomedial prelimbic, and hippocampal. All these sets included more than 40 artifact-free 2-s EEG epochs for each group (WT Vehicle, WT-BACE-1, TASTPM Vehicle, and TASTPM-BACE-1), behavioral condition (passive and active), and recording time (baseline and follow-up).

### 4.10. Spectral Analysis of the EEG Data

The artifact-free EEG epochs were used as an input for the spectral EEG data analysis, namely the computation of EEG power (density). This analysis was performed by a standard (MATLAB; MathWorks, Natick, Massachusetts USA) Fast Fourier Transform (FFT) algorithm using Welch technique and Hanning windowing (no overlap of the time windows) function with 0.5 Hz frequency resolution. For each mouse and electrode channel (frontoparietal, frontomedial prelimbic, and hippocampal), the normalization of the results of FFT analysis was obtained by computing the ratio between the absolute EEG power at each frequency bin and the absolute EEG power value obtained by the average across all frequency bins (0–100 Hz, frequency bin of net currents not considered), considering the two behavioral conditions (passive and active), and the two recording times (baseline and follow-up at 4 weeks of treatment). After this normalization, the EEG power lost the original physical dimension and was represented by an arbitrary unit scale. For example, in a certain mouse, if the maximum absolute EEG power is at 3 Hz with 150 µV^2^/Hz and the mean of the absolute EEG power across 0–100 Hz in the two conditions (active and passive) and two recording times (baseline and follow-up) is 15 µV^2^/Hz, then the normalized EEG power at 3 Hz is 10.

Two frequency bins of interest named as individual delta and theta frequencies (IDF and ITF), respectively, were detected. In each mouse, the IDF was defined as the frequency bin showing the highest amplitude of the normalized EEG power (density) between 1 and 6 Hz (typical extended delta frequency range) at the bipolar frontoparietal electrodes during the passive condition. Similarly, the ITF was defined as the frequency bin showing the highest amplitude of the normalized EEG power density between 6.5 and 10 Hz (typical extended theta frequency range) at the bipolar frontoparietal electrodes during the exploratory active condition. The frequency and amplitude of the IDF and ITF peaks were considered as markers of the normalized EEG power during the passive and the active condition, respectively. The same procedure was repeated for the intracerebral monopolar frontomedial prelimbic and hippocampal electrodes.

### 4.11. Statistical Analysis

Six statistical sessions were performed by the commercial tool STATISTICA 10 (StatSoft Inc., www.statsoft.com) to test the primary hypotheses of the present study by ANOVAs (*p* < 0.05). Mauchly’s test evaluated the sphericity assumption, and degrees of freedom were corrected by the Greenhouse–Geisser procedure when appropriate (*p* < 0.05). Finally, the Duncan test was used for post-hoc comparisons (*p* < 0.05), and the results were controlled by the Grubbs test (*p* < 0.01) for the presence of outliers. We used this statistical design based on ANOVAs and Grubbs test to make comparable the present findings with those of previous PharmaCog studies by our group applying the same methodology in WT [26], PDAPP [27] and TASTPM [28] mice.

In the first session, the ANOVA used EEG power (density) at the frontoparietal channel as a dependent variable to test the hypotheses that this variable may differ between the WT and TASTPM mice in relation to the behavioral conditions (passive vs. active) and be sensitive to the BACE-1 inhibitor over the vehicle treatment in the TASTPM mice (*p* < 0.05). The ANOVA used the normalized EEG power density as a dependent variable, while the factors were group (WT and TASTPM; independent variable), condition (active and passive), band (IDF, ITF, and beta), treatment (Vehicle and BACE-1; independent variable), and time (Day 0 and Week 4).

In the second and third session, the ANOVA and Grubbs test used EEG power (density) at the frontomedial and hippocampal channels, respectively, with the same structure and aim of the first session (*p* < 0.05).

In the fourth, fifth, and sixth sessions, the ANOVA and Grubbs test used the ratio between the normalized EEG power (density) of the ITF and IDF values (ITF/IDF) as a dependent variable to target the “reactivity” of EEG signals in the active over the passive condition for testing the above expected differences between the TASTPM Vehicle and BACE-1 inhibitor groups, and the effects of the treatment (*p* < 0.05). Specifically, the fourth ANOVA used this ratio for the frontoparietal channel. The ANOVA factors were group (WT and TASTPM; independent variable), condition (active and passive), treatment (Vehicle and BACE-1; independent variable), and time (Day 0 and Week 4). In the fifth and sixth sessions, the ANOVA and Grubbs test used that dependent variable for the frontomedial prelimbic and hippocampal electrodes, respectively (*p* < 0.05).

## 5. Conclusions

In the present study of the PharmaCog project (https://www.imi.europa.eu/projects-results/project-factsheets/pharma-cog), we tested whether ongoing EEG rhythms in wakefulness may reflect the effects of a chronic administration (4 weeks) of the anti-amyloid β-site APP-cleaving enzyme 1 inhibitor (BACE-1; ER-901356; Eisai Co, Ltd., Tokyo, Japan) in 8–9 month old TASTPM as compared to WT mice.

Results showed that in relation to the WT group, the TASTPM group generally showed a significantly lower “reactivity” in ongoing delta and theta rhythms during exploratory movements over a behavioral passive condition used as a reference. Notably, no effect of the BACE-1 inhibitor ER-901356 over vehicle administration was observed on those ongoing EEG rhythms; thus, suggesting that the present EEG markers may reflect differences in brain functions between the WT and TASTPM mice, but not the impact of a chronic treatment for 4 weeks of the present BACE-1 inhibitor on neocortical, limbic, and hippocampal neural synchronization mechanisms at the macroscale investigated in the present EEG study.

Future investigations may extend the present findings investigating the relationships among the present spectral EEG markers and the amyloid plaque formation, neuroinflammatory processes (e.g., by biomarkers of mRNA or protein expression), and cognitive performances in groups of at least 15 younger TASTPM mice (5 to 9 months of age) of both sexes for longer chronic administrations (i.e., several months) of the BACE-1 inhibitor ER-901356 in relation to vehicle and a control drug condition sensitive to AD neuropathology and EEG biomarkers (e.g., GABA-A agonist muscimol; [73]). Results of those future studies may allow to draw definitive conclusions on the tolerability and molecular, neurophysiological, and behavioral effects of the BACE-1 inhibitor ER-901356 treatment on the present spectral EEG biomarkers in this popular mouse model of AD-like amyloid neuropathology.

## Figures and Tables

**Figure 1 ijms-21-09072-f001:**
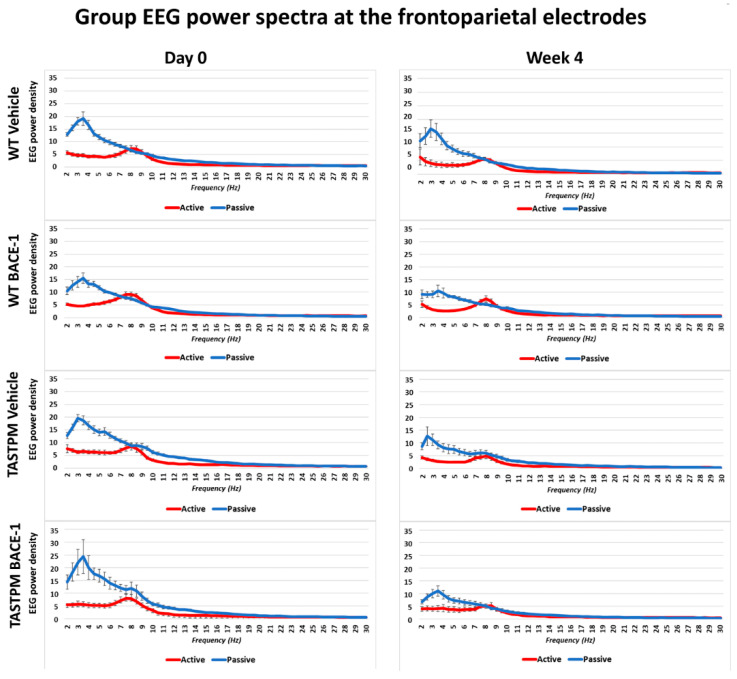
Mean values (± Standard Error of the Mean, SEM) of the normalized electroencephalographic (EEG) spectral power density at a frontoparietal (FP) recording channel for each frequency bin between 2 and 30 Hz (0.5 Hz resolution) during the active and passive behavioral conditions in the wakefulness for 4 groups of mice (wild type (WT) Vehicle, WT BACE-1, TASTPM Vehicle, TASTPM BACE-1). The EEG recordings were performed at the Day 0 and Week 4 of the administration of a BACE-1 inhibitor (ER-901356; Eisai Co, Ltd., Tokyo, Japan) or a dose of vehicle through tube feeding.

**Figure 2 ijms-21-09072-f002:**
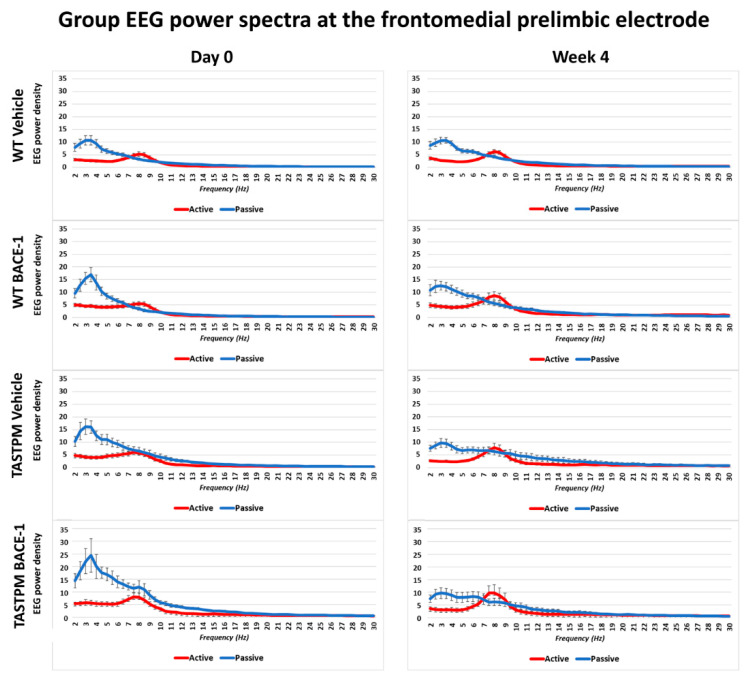
Mean values (± SEM) of the normalized EEG spectral power density at a frontomedial prelimbic (PLC) recording channel for each frequency bin between 2 and 30 Hz (0.5 Hz resolution) during the active and passive behavioral conditions in the wakefulness for each group of mice (WT Vehicle, WT BACE-1, TASTPM Vehicle, TASTPM BACE-1). The EEG recordings were performed at day 0 and week 4.

**Figure 3 ijms-21-09072-f003:**
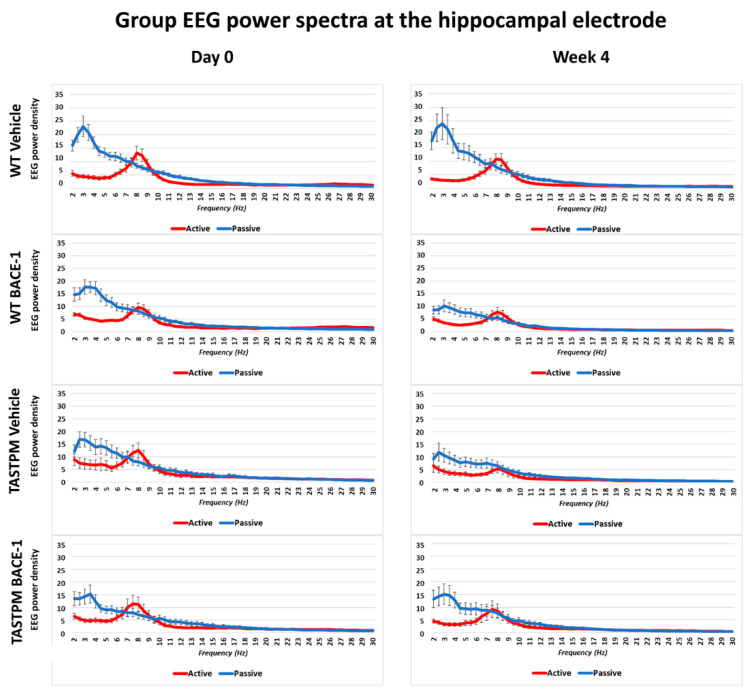
Mean values (± SEM) of the normalized EEG spectral power density at a hippocampal (Hipp) recording channel for each frequency bin between 2 and 30 Hz (0.5 Hz resolution) during the active and passive behavioral conditions in the wakefulness for each group of mice (WT Vehicle, WT BACE-1, TASTPM Vehicle, TASTPM BACE-1). The EEG recordings were performed at day 0 and week 4.

**Figure 4 ijms-21-09072-f004:**
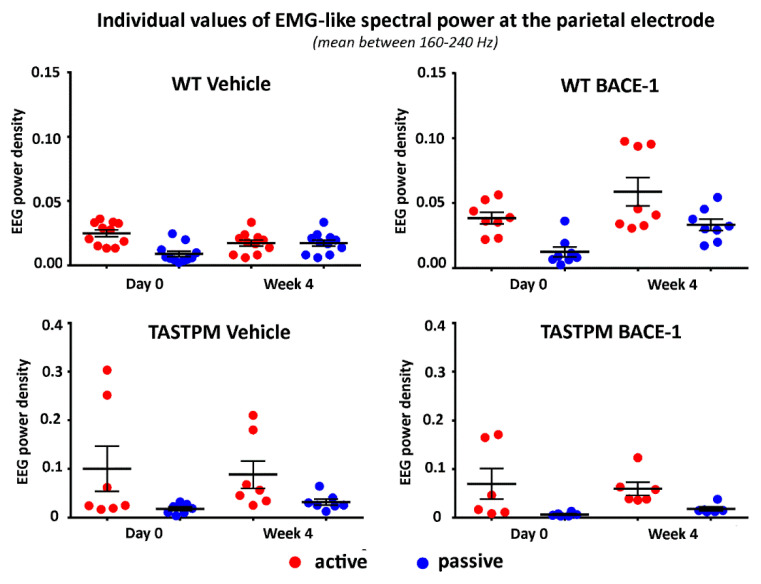
Individual values (± SEM) of the normalized spectral power density (mean between 160 and 240 Hz) of the electromyographic-like activity recorded at a parietal exploring electrode during the active and passive behavioral conditions in the wakefulness for each group of mice (WT Vehicle, WT BACE-1, TASTPM Vehicle, TASTPM BACE-1) at day 0 and week 4. Such an electromyographic-like activity recorded at the parietal electrode is supposed to be generated mainly from neck and dorsal muscles of the mice. Two outlier values in the WT BACE-1 group and 1 outlier value in the TASTPM Vehicle group were excluded according to a Grubbs test (*p* < 0.01).

**Figure 5 ijms-21-09072-f005:**
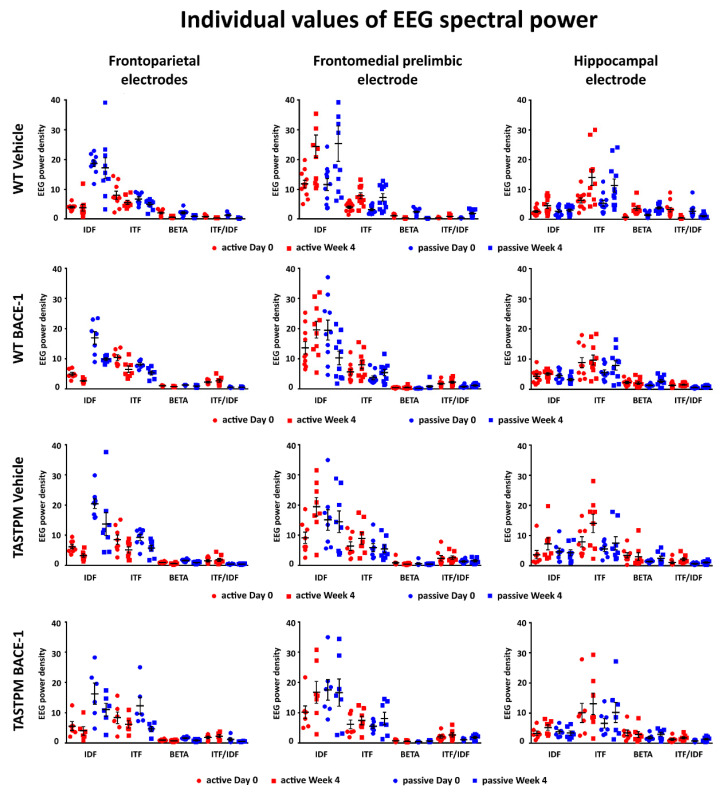
Individual values (± SEM) of the normalized EEG spectral power density at the frontoparietal (FP), frontomedial prelimbic (PLC), and hippocampal (Hipp) recording channels during the active and passive conditions in the wakefulness for each group of mice (WT Vehicle, WT BACE-1, TASTPM Vehicle, TASTPM BACE-1). The EEG recordings were performed at day 0 and week 4. No outlier values were observed according to a Grubbs test (*p* < 0.01).

**Figure 6 ijms-21-09072-f006:**
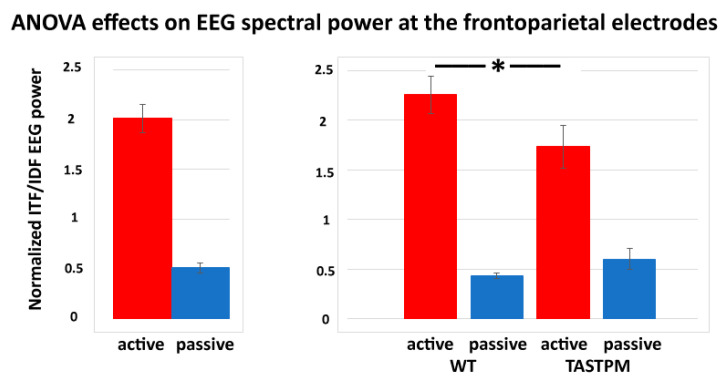
Mean values (± SEM) of the normalized individual delta and theta frequency peaks (ITF/IDF) EEG spectral power density at the frontoparietal (FP) recording channel during the active and passive conditions in the wakefulness for each group of mice (WT Vehicle, WT BACE-1, TASTPM Vehicle, TASTPM BACE-1). The EEG recordings were performed at day 0 and week 4. The diagram on the left illustrates the main effect of the condition factor (F (1, 52) = 84.1, *p* < 0.001), while the diagram on the right illustrates the ANOVA interaction between the condition (active, passive) and group (WT, TASTPM) factors [F (1, 52) = 5.0, *p* < 0.05]. The original ANOVA design had the normalized ITF/IDF EEG power density at the frontoparietal (FP) recording channel as a dependent variable and the condition (active, passive), group (WT, TASTPM), treatment (Vehicle, BACE-1), and time (day 0, week 4) as factors. The statistically significant difference between the WT and TASTPM mice is indicated by the asterisk (Duncan’s post-hoc test, * = *p* < 0.05).

**Figure 7 ijms-21-09072-f007:**
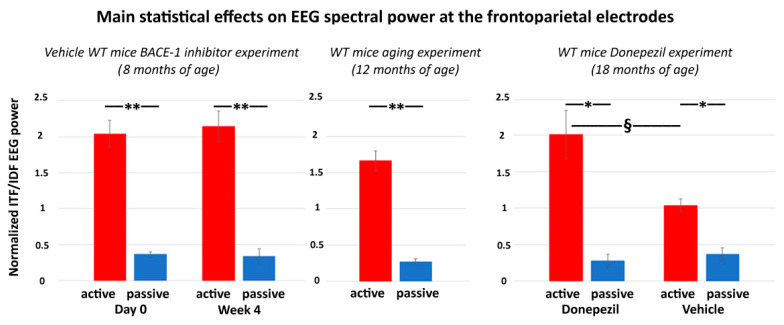
Mean values (± SEM) of the normalized ITF/IDF EEG spectral power density at the (bipolar) frontoparietal (FP) derivation during the active and passive conditions recorded in the wakefulness for each group of WT mice used in the control experiment. The middle diagram illustrates the difference of the normalized ITF/IDF EEG spectral power density at the bipolar frontoparietal derivation between the active and passive conditions in the WT mice 12 months old receiving no intervention with vehicle or Donepezil (Wilcoxon test, z = 2.67, *p* < 0.01). The right diagram illustrates the results obtained in the WT mice 18 months old receiving a single dose of vehicle or Donepezil. There are differences in the normalized ITF/IDF EEG spectral power density at the bipolar frontoparietal derivation (1) between the active and passive conditions after both vehicle and Donepezil interventions (Wilcoxon test, z = 2.67, *p* < 0.01) and (2) between the vehicle and Donepezil interventions for the active condition (Mann–Whitney U test, z = 2.09, *p* < 0.05). For illustrative purposes, the left diagram illustrates the difference of the normalized ITF/IDF EEG spectral power density at the bipolar frontoparietal derivation between the active and passive conditions in the WT Vehicle mice 8 months old involved in the main experiment (BACE-1 inhibitor vs. vehicle) at both day 0 and week 4 (Wilcoxon test, z = 2.67, *p* < 0.01). Legend: * = active ≠ passive, *p* < 0.05; ** = active ≠ passive, *p* < 0.01; § = Vehicle ≠ Donepezil, *p* < 0.05.

**Table 1 ijms-21-09072-t001:** Features of the C57BL/6 (for the sake of simplicity, wild type, WT) and the transgenic TASTPM mice undergoing to a chronic administration lasting 4 weeks of a dose of 10 mg/kg (milligram per kilogram) of a β-site APP-cleaving enzyme 1 (BACE-1) inhibitor (ER-901356; Eisai Co., Ltd., Tokyo, Japan) or a dose of vehicle through tube feeding. In the present experiments, those mice underwent ongoing electroencephalographic (EEG) recordings at day 0 and week 4 of the administration.

Genotype	Treatment	*n* (At Day 0)	*n* (At Week 4)	Age	Gender
*WT*	Vehicle	11	11	8–9 months	Males
*WT*	BACE-1 inhibitor	10	10	8–9 months	Males
*TASTPM*	Vehicle	10	8	8–9 months	Males
*TASTPM*	BACE-1 inhibitor	11	7	8–9 months	Males

**Table 2 ijms-21-09072-t002:** Time flow of present experimental procedures considering the days in relation to the surgical event. EEG = Electroencephalography; mpg = mg /kg of body weight; p.o. = oral administration.

Period	Days	Treatment and Procedures
Pre-surgery	−21 to −1	✓ Habituation to light switched on–off ✓ Gentle handling for about 5–10 min a day
Surgery	0	✓ Anesthetic procedure ✓ Therapy with systemic analgesics and antibiotics ✓ Electrode placement
Quiet post-surgery	+1 to +7	✓ Therapy with systemic analgesics and antibiotics ✓ No gentle handling ✓ No EEG experiment
Post-surgery	+8 to +14	✓ Facilitating the adaptation by plugging and unplugging several times the animal ✓ Gentle handling for about 5–10 min a day ✓ No EEG experiment
Baseline (Day 0) EEG recording	+15	✓ No gentle handling ✓ EEG recording
Drug administration	from +15 to 43	✓ Vehicle or drug (BACE-1 inhibitor ER-901356, Eisai) administration 10 mpk p.o. daily (tube feeding)
Follow-up (Week 4) EEG recording	+43	✓ No gentle handling ✓ EEG recording

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
