# Peer review of "Chronic BACE-1 Inhibitor Administration in TASTPM Mice (APP KM670/671NL and PSEN1 M146V Mutation): An EEG Study"

_ijms, 2020, doi:10.3390/ijms21239072_

Round 1
Reviewer 1 Report
The study of Babiloni et al deals with the effect of Bace1 inhibition in TASTPM mice as an Alzheimer model and wt mice on delta and theta frequencies in EEG rhythm. Main findings were a decreased IFT power in TASTPM compared to wt mice but no statistically significant effects were observed for chronic (4 weeks) treatment with Bace 1 inhibitor.
This study is interesting for a relative broad readership, especially for scientists dealing with Alzheimer´s disease or EEG studies in general. It has a clear focus and a straight forward design and is clearly written. However, I have especially some principal and major concerns, which makes - in my opinion - the interpretation of the results complicated.
Principal and major concerns:
- The authors utilize TASTPM mice for their study carrying two mutations associated with early onset disease: the Swedish mutation in APP and the M146V mutation in PSEN1. They develop several AD-related phenotypes, including age-related amyloid pathology and memory deficits, but they do not - or to a very low extend - develop tangle pathology or appreciable neuronal loss. Tangle pathology seems to be highly relevant in EEG measurements. E.g. a very recent study has shown that there is a hyperexcitability in the THY-Tau22 mouse model of tauopathy (Neurobiol Aging. 2020 Jun 20;94:265-270). Therefore, the mouse model the authors use here focuses the Amyloid cascade neglecting the tau pathology in AD. At least this issue should be very carefully discussed.
- In line with the above mentioned point the authors use a mouse model with the APP Swedish mutation. To my knowledge the APP Swedish mutation increases the affinity of APP to BACE1 and therefore increases the amyloidogenic pathway. I am not sure if the Bace1 inhibitor under the conditions the authors use here is able to compensate the APPswe effect. As a proof of concept, the authors cite a study (line 520ff, Chang 2010), in which APPswe transgenic mice treated with a Bace1 inhibitor showed a reduced accumulation of Abeta in the brain. However, the mice were here continuously infused with the inhibitor for 6 months. So, I really wonder, if the lack of an effect is only due to a very small effect of the inhibitor. This is an essential point. To make any conclusion the author should provide UNDER THEIR EXPERIMENTAL CONDITIONS data showing how plaque burden, Bace activity, Abeta level etc. were affected. Without these data it is impossible to judge if the lack of an effect is due to a lack on Abeta homeostasis or if Abeta homeostasis has only a minor effect on EEG measurements. So please provide some biochemical data reflecting exactly your experimental conditions which can be correlated with the EEG findings.
Minor concerns
- More as a comment: In TASPM mice a gender effect is observed. Female mice develop more rapid and severe amyloid pathology than males, with a greater Aβ load at all ages examined. I wonder why the authors decided to use male mice.
- 4 mice (out of 11) in the treated group died (2 fold more as in the vehicle group) suggesting to be very careful with the interpretation of the results because the treatment does not see to be tolerated very well.
- Another point is if the statistical power is high enough, taken into consideration that he n number is reduced by 30% because the mice died.
Reviewer 2 Report
I think the authors present an interesting work. However it is still too exploratory, needing more animals to clear some of the claims and above all the message is not clear, since the drug treatment they use is a negative control, meaning was not expected to ameliorate the transgenic Alzheimer mice. So it is difficult to talk about EEG biomarkers, explored in this paper.
- Main findings of the study:
. The authors evaluated how electroencephalographic (EEG) rhythms, previously shown to discriminate WT from Alzheimer transgenic mice (TASTPM), can be used to test Alzheimer therapeutics, such as BACE-1 (anti-amyloid β-site APP-cleaving enzyme 1 inhibitor)
.The authors followed the EEG rhythms of WT/TASTPM mice with/without BACE-1 chronic administration for 4 weeks.
.Authors registered the frequency and magnitude of individual EEG delta and theta frequency peaks. TASTPM mice showed lower theta frequency peaks. However, no BACE-1 inhibitor effect over vehicle was observed.
. Authors claim that EEG biomarkers reflect differences between WT and TASTPM mice (already known) but not of in the 4-week treatment of BACE-1.
- Limitations:
. I think the abstract should be more clear on the results obtained. Moreover some important results are missing, such as the different EECG rythms analyses in the different parts of the brain (describe the method, not the results). Moreover it is not clear if the treatment described BACE-1 inhibitor, in the conditions tested ( 9 month old TASTPM mice, 1 month administration) clearly improves brain amyloidosis and cognitive deficits in these mice! In discussion it is referred that this treatment did not show any beneficial effects…so why use it? No positive control?
. BACE-1 inhibitor is affecting EEG rhythms in WT mice after 4 weeks, at least in the passive condition, so it is difficult to understand TASTPM result’s BACE-1 results. In all the different areas of the brain. So, combined graphics would help to understand the differences (in the frequencies where differences are observed (2-12 Hz)
. The number of animals for this kind of analysis, behavioral, should be around 20, to have more robust data and statistical power. Although the number used is reasonable.
. Authors used an anesthetic (chloral hydrate) that has been completely outdated and even advised against, being substituted by better compounds (in terms of surgical refinement), like medetomidine/ketamine, isoflurane, etc. This drug is a sedative and for this reason does not completely prevent suffering in animals.
Major revisions:
. Authors should follow the ARRIVE guidelines for reporting of in-vivo animal experiments. So important recommendations are missing such as; a time-line or flow chart illustrating the study; post-surgery animal care (pain relief?); exclusion criteria; randomization; etc. However, a good description is already present.
. As described by the authors, this is an exploratory study, where further animals should be used and some other treatment should be issued that reduces amyloid and improves behavior to understand if EEG can be used as a potential biomarker. Authors could use for example Epigallocatechin-3-Gallate (EGCG).
. Authors should either focus on the differences between WT and Alzheimer model or if they want to analyze treatments they should have an effective treatment…that actually reduces amyloid and ameliorates cognitive behavior, otherwise the paper is not conclusive and is unfinished regarding EEG as potencial biomarkers!
Reviewer 3 Report
The authors have done a fine job of attempting to explore the differences in EEG signals between animals for WT and TASTPM subgroups. The hypothesis of this study is interesting.
I have a number of comments :
Title : TASTPM, "mutation" not addressed
The content of the introduction is redundant.
In animal experiments, differences in experimental data tend to be observed in the same group. The inability to draw more convincing conclusions may be related to the lack of numbers of subjectives in each group
Line 602-5. The content of this part should be moved to the limitation of the experiment and should be discussed more extensively.
As the author indeed mentioned, the molecular signaling of inflammation or potential underlying pathological mechanisms , such as results of mRNA or protein expression should be added to the results. If not, an explanation in detail should be addressed in the discussion.
A professional English proofreading is suggested.
Round 2
Reviewer 1 Report
The authors have sufficiently addressed all my concerns and remarks. In my opinion the manuscript is suitable for publication in its present form.
Author Response
We thank the Reviewer #1 for his/her final comments and acceptance of the revised manuscript.
Reviewer 2 Report
I acknowledge the work done in the revision performed by the authors, however I think for the work to be relevant for others, some of the limitations and major revisions pointed should be performed. Such as the use of an effective control drug (as you mention in the abstract now), otherwise this could not be even be considered a negative result publication, in my view.
So, I think the authors should complete the study so that it can be more clear and significant.
Reviewer 3 Report
All the points raised have been addressed in the revised version.
Author Response
We thank the Reviewer #3 for his/her comments.